# Estimation of the Overuse of Preoperative Chest X-rays According to “Choosing Wisely”, “No Hacer”, and “Essencial” Initiatives: Are They Equally Applicable and Comparable?

**DOI:** 10.3390/ijerph17238783

**Published:** 2020-11-26

**Authors:** Jorge Vicente-Guijarro, José Lorenzo Valencia-Martín, Paloma Moreno-Nunez, Pedro Ruiz-López, José Joaquín Mira-Solves, Jesús María Aranaz-Andrés

**Affiliations:** 1Servicio de Medicina Preventiva y Salud Pública, Hospital Universitario Ramón y Cajal, IRYCIS, 28034 Madrid, Spain; pmorenon@salud.madrid.org (P.M.-N.); jesusmaria.aranaz@salud.madrid.org (J.M.A.-A.); 2Departamento de Medicina y Especialidades Médicas, Facultad de Medicina, Universidad de Alcalá, 28801 Acalá de Henares, Madrid, Spain; 3Instituto Ramón y Cajal de Investigación Sanitaria, IRYCIS, 28034 Madrid, Spain; jose.valencia@salud.madrid.org; 4Servicio de Medicina Preventiva y Salud Pública, Unidad de Gestión Clínica de Prevención, Promoción y Vigilancia de la Salud, Hospital La Merced, Área de Gestión Sanitaria de Osuna, 41640 Osuna, Sevilla, Spain; 5Facultad de Ciencias de la Salud, Universidad Internacional de la Rioja, 26006 Logroño, La Rioja, Spain; pruizl@salud.madrid.org; 6Hospital Universitario 12 de Octubre, 28041 Madrid, Spain; 7Health Psychology Department, Miguel Hernández University, 03202 Elche, Spain; jose.mira@umh.es; 8Alicante-Sant Joan Health District, Ministry of Health, 03550 Alicante, Spain; 9REDISSEC, Health Services Network Oriented to Chronic Diseases, Spain; 10CIBER Epidemiología y Salud Pública (CIBERESP), 28034 Madrid, Spain

**Keywords:** medical overuse, practice guidelines as topic, preoperative care, diagnostic tests, routine, anesthesia department, hospital

## Abstract

Background: Overuse reduces the efficiency of healthcare systems and compromises patient safety. Different institutions have issued recommendations on the indication of preoperative chest X-rays, but the degree of compliance with these recommendations is unknown. This study investigates the frequency and characteristics of the inappropriateness of this practice. Methods: This is a descriptive observational study with analytical components, performed in a tertiary hospital in the Community of Madrid (Spain) between July 2018 and June 2019. The inappropriateness of preoperative chest X-ray tests was analyzed according to “Choosing Wisely”, “No Hacer” and “Essencial” initiatives and the cost associated with this practice was estimated in Relative Value and Monetary Units. Results: A total of 3449 preoperative chest X-ray tests were performed during the period of study. In total, 5.4% of them were unjustified according to the “No Hacer” recommendation and 73.3% according to “Choosing Wisely” and “Essencial” criteria, which would be equivalent to 5.6% and 11.8% of the interventions in which this test was unnecessary, respectively. One or more preoperative chest X-ray(s) were indicated in more than 20% of the interventions in which another chest X-ray had already been performed in the previous 3 months. A higher inappropriateness score was also recorded for interventions with an American Society of Anesthesiologists (ASA) grade ≥ III (16.5%). The Anesthesiology service obtained a lower inappropriateness score than other Petitioning Surgical Services (57.5% according to “Choosing Wisely” and “Essencial”; 4.1% according to “No Hacer”). Inappropriate indication of chest X-rays represents an annual cost of EUR 52,122.69 (170.1 Relative Value Units) according to “No Hacer” and EUR 3895.29 (2276.1 Relative Value Units) according to “Choosing Wisely” or “Essencial” criteria. Conclusions: There was wide variability between the recommendations that directly affected the degree of inappropriateness found, with the main reasons for inappropriateness being duplication of preoperative chest X-rays and the lack of consideration of the particularities of thoracic interventions. This inappropriateness implies a significant expense according to the applicable recommendations and therefore a high opportunity cost.

## 1. Introduction

In 1998, Mark R. Chassin and Robert W. Galvin defined the overuse of the healthcare system as any process that carries more risks than benefits for the patient [1]. This public health problem implies the performance of unjustified procedures on the patient which, from a safety perspective, means making them assume unnecessary health risks and, economically, entails an opportunity cost.

Overuse has been investigated incrementally over the last few years [2] and has been addressed from the perspective of health professionals themselves [3,4,5,6], distribution by geographical areas [7] and economic repercussions [8]. Various methodologies have also been used for studies, such as the analysis of claims filed with the healthcare system [9,10] or through systematic reviews, estimating that approximately 40% of current procedures did not provide greater care value than the previously existing clinical alternative [11].

In 2012, “The American Board of Internal Medicine” (ABIM) Foundation promoted the project “Choosing Wisely” [12] (CW), which was the campaign to reduce overuse that had the greatest international impact to date. The National Institute for Health and Care Excellence (NICE) identified by consensus a set of practices that did not add value or that could even be harmful and included them in the “Do Not Do” (DND) list [13].

In Spain, the “Ministry of Health, Social Services and Equality” (MSSSI) promoted the initiative “Commitment to the Quality of Scientific Societies” (ICC) [14] in 2013. Applying a methodology similar to that used by the CW initiative, more than 40 Spanish scientific societies in the healthcare field chose the five practices of each medical specialty that did not provide clinical benefit and compiled them in the “No Hacer” (NH) (Spanish equivalent to Do Not Do) initiative. Similarly, the Regional Government of Catalonia developed “Essencial”, a similar project promoted by the Catalan Agency for Quality and Health Assessment (AQuAS) [15].

A potentially unnecessary clinical practice, for which all previous institutions have developed a recommendation, was the unjustified indication of chest X-rays (CXRs) as part of a preoperative anesthetic study [16,17,18,19]. Several studies suggest that the value of this test would be limited, due to the low percentage of X-rays in which relevant findings are found [20,21] and because its performance would rarely influence the postoperative management of the patient [22,23,24,25].

Furthermore, this practice can have negative consequences on patient safety. First, because each CXR emits an average cumulative radiation dose of 0.02 milliSievert [26]; second, due to the intrinsic risk of finding “false positive” findings [27]; third, because these “false positives” could trigger cascading requests for other, also unjustified, procedures posing even more risks for the patient [28,29].

Some studies have evaluated these costs in different healthcare systems. In 2018, The Washington Health Alliance estimated that, collectively, 20% of electrocardiograms, preoperative chest X-rays (PCXRs) and pulmonary function tests that had been performed in the State of Washington (United States) over a year, were not justified, which would represent an unnecessary expense of USD 6.4 million [30]. In Spain, an improvement campaign carried out in the Community of Madrid between 2010 and 2014 estimated that more than one million unnecessary CXRs could have been avoided, with a saving of more than EUR 412,000 for that period [26].

In Spain, to date, the frequency of overuse of PCXRs in specialized healthcare has not been investigated. This study aimed to estimate the degree of inappropriateness of PCXRs in a tertiary hospital, considering the different recommendations issued by international scientific institutions, as well as analyze characteristics associated with the unnecessarily indicated PCXRs.

## 2. Materials and Methods

### 2.1. Study Design

This is a descriptive observational study with analytical components performed through the information system records of a tertiary hospital in the Community of Madrid (Spain) with a capacity of 901 beds and 45 operating rooms. The study population comprised the interventions performed between 1 July 2018 and 30 June 2019, as well as the indications for CXRs associated with their respective clinical episodes. These were the only inclusion criteria. The exclusion criterion was if the associated CXRs had been performed after the intervention.

All the data were categorized (except for the reason for the indication of the CXR, which was recorded by each requesting physician in a free text field) and were extracted by the Information Systems Unit of the Admission Coordination, Care and Clinical Documentation Information Systems of the hospital.

### 2.2. Selected Recommendations

The degree of inappropriateness of the indication of PCXRs was studied based on the following recommendations:

“The ABIM Foundation”, in collaboration with “The American College of Radiology” (ACR), established the following recommendation through the CW initiative: “If you do not have symptoms of heart or lung disease, and your risk is low, an X-ray probably will not help (…) It is a good idea to have a chest X-ray before you have surgery or before you go into the hospital if (1) You have signs or symptoms of a heart or lung condition. These include chest pain, coughing, shortness of breath, swelling in the ankles, fever, a recent heart attack, or a cold or other lung infection that does not go away. (2) You have heart or lung disease, whether or not you have symptoms. (3) You are older than 70 and you have not had a chest X-ray within the last six months. (4) You are having surgery on the heart, lungs, or any other part of the chest” [16].

The MSSSI, in collaboration with the “Spanish Society of Anaesthesiology, Resuscitation and Pain Therapeutics” (SEDAR), included “The American Society of Anesthesiologists (ASA)” physical status classification system [31] in its evaluation, and established the following recommendation through the NH initiative: “do not perform preoperative chest radiography in patients under 40 years of age with physical status ASA I or II” [18]. In this respect, The ASA physical status classification system includes “healthy patients” as ASA I and “patients with mild systemic disease” as ASA II, while an ASA grade ≥ III implies a “severe systemic disease” or worse health conditions [31].

The AQuAS of the “Regional Government of Catalonia”, in collaboration with the “Catalan Society of Anaesthesiology, Resuscitation and Pain Therapeutics” (SCARTD) and with “Radiologists of Catalonia”, established the following recommendation through the “Essencial” initiative: “scientific evidence shows that routine chest X-rays before a surgical intervention (preoperative) in patients without symptoms of heart or lung disease does not lead to any improvement in their clinical management. (…) Preoperative chest X-ray would be indicated in thoracic surgery, in patients who present signs or symptoms of heart or lung disease, and in patients older than 70 years with stable chronic cardiopulmonary disease and more than six months have passed since the last X-ray” [19].

Thus, CW and “Essencial” initiatives can be considered very similar recommendations that barely differ in small details such as (1) CW specifies the possible symptoms of heart or lung condition. (2) CW considers the PCXR indicated in patients with heart or lung condition appropriate, regardless of the presence of symptoms. (3) “Essencial” indicates to order PCXRs on patients older than 70, with stable chronic cardiopulmonary disease, and without previous CXR in the last 6 months; while CW does not consider necessary the existence of stable chronic cardiopulmonary disease for this indication.

### 2.3. Analysis Plan

Classification algorithms were developed for the interpretation of CXRs, such as “preoperative CXR” (PCXR) and “non-pre-operative CXR”, depending on the reason for the indication and the previous surgical interventions, classifying PCXRs as those in which this indication was unequivocally recorded and were performed prior to surgery.

The records did not provide information about chronic disease, signs, or symptoms. For this reason, to assess inappropriateness according to the recommendations of the CW and “Essencial” initiatives, it was considered that those patients whose intervention had a Diagnosis Related Group (DRG) code of surgery that compromised the thoracic cavity had signs, symptoms, or a diagnosis of heart or pulmonary disease and they would have been indicated for PCXR correctly. However, it was not possible to identify patients with these conditions but without a thoracic cavity surgery DRG code. Therefore, the quality of the records did not allow us to explore the subtle differences existing between CW and “Essencial” regarding the presence of chronic disease, signs, or symptoms. Thus, both recommendations were finally assessed under the same criteria, and their results are shown together in the next sections of the manuscript.

To assess the existence of other previous CXRs, other CXRs performed in the same hospital or peripheral specialty centers 3, 6, 9 and 12 months prior to the intervention were identified.

The percentage of unjustifiably indicated PCXRs and the percentage of interventions in which at least one PCXR was indicated, of the total number of interventions in which this indication was unnecessary, was assessed based on combinations of the following assumptions of inappropriateness: ASA grade I or ASA grade II, age under 40, 60 or 70 years at the time of surgery, DRG not related to an intervention on the thoracic cavity and performance of other CXR in the 3, 6, 9 or 12 months prior to the intervention. The inappropriateness assumptions of the CW, NH and “Essencial” recommendations were included among these combinations (Figure 1). Additionally, both values were assessed based on the other available variables.

A descriptive analysis was performed by calculating the frequency estimators (percentages, means and standard deviation) with respective 95% confidence intervals (95% CIs). The analysis was performed globally and stratified into “Inappropriate preoperative CXRs” (iPCXRs) and “Appropriate preoperative CXRs”.

To estimate the healthcare cost corresponding to the iPCXR, Relative Value Units (RVUs) were used as a measuring tool that integrates the personnel cost, maintenance cost, depreciation costs and other charges directly related to the performance of a healthcare procedure [32]. Thus, this study used the value of 0.9 RVU per iPCXR, stipulated by the Community of Madrid [33] and the “Spanish Society of Medical Radiology” [34]. To calculate the equivalence in monetary units, the equivalence of 1 RVU was set at EUR 22.90, established by the “Health Service of the Basque Country” (Osakidetza) for 2020 [35], as this is the most up-to-date approximation available and closest to the scope of the study.

### 2.4. Bivariate Analysis

The possible association between the inappropriateness in the indication of PCXRs was assessed with the following variables: (1) Related to the patient: gender, age and ASA grade [31] at the time of the intervention. (2) Related to the intervention: Care Service Responsible, “urgent” or “scheduled” nature, DRG, existence of another intervention in the previous 12 months and existence of another chest X-ray in the previous 3, 6, 9 or 12 months. (3) Related to the PCXR: requesting department.

For the comparison of qualitative variables, the parametric Chi-square test (χ^2^) was used. In the case of failure to comply with the requirements for applicability of this parametric test, Fisher’s exact test was used. The 95% confidence intervals (α = 0.05) and significance *p*-value were estimated for all frequency estimates. Those differences with a *p*-value less than 0.05 were considered statistically significant. The statistical treatment of the data and the free text field recoding algorithms were developed and executed using the statistical software Stata^®^ v.13 (College Station, TX, USA) [36].

### 2.5. Ethical Statement

This project obtained the approval of the Management of the Hospital Universitario Ramón y Cajal (Madrid, Spain), and a favorable opinion by the research ethics committee of the same center (reference 168/17) dated 31 July 2017 (31/07/2017).

## 3. Results

Throughout the year studied, 27,890 interventions were performed in the hospital and 3449 PCXRs were indicated. In 10.4% (2912) of the interventions, at least one PCXR was indicated, with a mean of 0.1 PCXR indicated per intervention. A total of 52.2% of the interventions were performed on women and the median age was 64 years (quartiles 1 and 3: 48 and 76 years). The following aspects accounted for a higher percentage of “interventions with PCXR” over “interventions without PCXR”: male patients, ≥40 years old, with an ASA grade ≥ III, looked after by the Departments of Traumatology, Urology, Cardiac Surgery and Thoracic Surgery; an associated DRG that involved the thoracic cavity, with scheduled operations, or with the presence of another CXR 3, 6, 9 or 12 months prior to the intervention (Table 1).

Of the 3449 PCXRs indicated, the Surgical Departments that requested them most frequently were Traumatology (15.3%; 529) and General Surgery (9.9%; 343). In total, 32.0% (1105) of the PCXRs were indicated by department other than the one to which the intervention was associated, with Anesthesiology (44.1%; 487) and Emergencies (11.4%; 126) being the departments making requests most frequently.

### Inappropriateness of PCXR Indication

Of the total number of PCXRs requested (3449), 73.3% (2529) were iPCXRs according to the CW and “Essencial” initiatives, and 5.4% (187) according to *NH*. Additionally, the percentage of interventions in which at least one PCXR was indicated, of the total number of interventions in which this indication was unnecessary, was 11.8% (2257) according to the CW and “Essencial” initiatives and 5.6% (174) according to *NH* (Table 2 and Table 3).

According to all the recommendations evaluated, more iPCXRs were indicated in interventions in which another X-ray had been performed in the previous 3, 6, 9 or 12 months compared to those who did not present such a medical history (Table 4).

On the other hand, according to CW and “Essencial”, more iPCXRs were indicated in “urgent” interventions (14.7%, compared to 11.6% in “non-urgent”; *p* = 0.001), this trend reversed with the application of the NH criteria (6.1% in “non-urgent”, compared to 2.2% in “urgent”; *p* = 0.002).

According to CW and “Essencial”, more iPCXRs were indicated in interventions with patients with an ASA grade ≥ III (16.5%, compared to 10.5% in ASA I-II; *p* < 0.001); in patients ≥40 years (13.4%, versus 5.5% in patients <40; *p* < 0.001), in patients ≥60 years (15.0%, versus 9.0% in patients <60; *p* < 0.001) and in patients ≥70 years (17.7%, compared to 10.0% in <70; *p* < 0.001).

On the other hand, according to the NH initiative, more iPCXRs were indicated in interventions with thoracic DRG (18.6% versus 5.7% in other DRGs; *p* < 0.001).

According to all the recommendations assessed, Anesthesiology indicated fewer iPCXRs than other services such as Maxillofacial Surgery, Otorhinolaryngology or Ophthalmology (*p* < 0.001). Cardiac Surgery and Thoracic Surgery also presented a lower degree of inappropriateness than the others, according to the CW initiative (*p* < 0.001) (Table 5).

The cost associated with iPCXR was 2276.1 RVU/year according to CW or “Essencial” and 170.1 RVU/year according to NH, which would represent 52,122.69 and 3895.29 EUR/year, respectively.

## 4. Discussion

Wide variations were found in the degree of inappropriateness of PCXRs according to the recommendations assessed (5.4% according to “No Hacer” recommendation, and 73.3% according to “Choosing Wisely” and “Essencial” criteria). The performance of another CXR in the months prior to the intervention showed a high association for inappropriateness of PCXR, although the young age of the patient and the absence of a thoracic cavity surgery DRG would also increase overuse. Unnecessary PCXRs were indicated more in ASA grade ≥ III patients according to CW and “Essencial”; the Services of Maxillofacial Surgery, Otorhinolaryngology, or Ophthalmology also indicated more unnecessary PCXR than Cardiac Surgery, Thoracic Surgery, or Anesthesiology. The cost associated with overuse shows a high variability between 52,122.69 and 3895.29 EUR/year, depending on the recommendation applied.

There were wide variations in the degree of inappropriateness of PCXRs obtained depending on the recommendation applied; inappropriateness being 14 times higher according to CW and “Essencial” compared to NH. The variation was slightly lower than that obtained in Canada in 2004, where a PCXR inappropriateness of 15.9%, 54.0% and 98.4% was estimated according to the guidelines of “The Ontario Preoperative Task Force”, “The Canadian Anesthesiologists’ Society” and “The Ottawa Hospital”, respectively [37]. The results obtained were also lower than those observed in Austria in 2007 (84%), according to a guide from “The Austrian Society of Anaesthesiology, Resuscitation and Intensive Care Medicine” [38]. However, these publications applied other criteria, had a smaller sample size than that used in our study (63 and 410, compared to the 3449 PCXRs assessed here) and were carried out in Health Systems other than Spanish ones.

Performing another preoperative CXR in the months prior to the intervention constitutes the assumption associated with a greater increase in inappropriateness. For example, if the presence of another CXR was added in the previous 6 months as a criterion of inappropriateness according to the NH recommendation, it would go from an inappropriateness of 5.4% of the PCXRs to 92.4%, or from 5.6% to 17.5% of the total number of interventions in which such an indication was unnecessary. Although there is no clear consensus on what the validity period of a PCXR, CW and “Essencial” is, it is set at 6 months for patients older than 70 years [16,19]; The Royal College of Radiologists, in a guideline from 1994, extended it up to 12 months in smokers or patients with cardiorespiratory disease [39]. This period of validity for PCXRs would be consistent with the results obtained in a study carried out in 2017 in Brazil, which concluded that the probability of finding changes in the year after a normal result in a preoperative test would be only 1.7% [40]. On the other hand, the results of our study for CW and “Essencial” are congruent with those of another study performed in Austria in 2007, in which it was found that up to 80% of duplicate PCXRs could be unnecessary [38]. However, duplication of tests has been more widely studied in the field of pharmacological prescription, where it was estimated that up to 76.4% of duplications would have their origin in prescriptions indicated by different professionals [41].

The consideration of a thoracic intervention as a criterion for the correct indication of a PCXR supposes a significant decrease in the resulting inappropriateness. The higher percentage of iPCXRs obtained in interventions with thoracic DRG according to the NH initiative (18.6% compared to 5.7% in other DRGs), could be explained if the requesting professionals had considered it necessary to indicate a PCXR in case of intervention or pathology thoracic cavity (as dictated by CW and “Essencial”, but not NH). This could also have influenced the greater degree of inappropriateness obtained between the General Surgery, Otorhinolaryngology or Ophthalmology Departments, compared to Anesthesiology (and with Cardiac Surgery and Thoracic Surgery, according to CW). However, when assessing the degree of inappropriateness of PCXRs based on the different requesting services, in general, high levels were obtained according to CW and “Essencial” and discrete according to NH criteria. Although this finding is consistent with the variability in the indication of PCXRs between departments observed in the United Kingdom in 1979 [42], the general lack of studies that assess these differences considerably hinders comparison.

Regarding the ASA classification, although more iPCXRs were indicated in ASA grade ≥ III patients according to CW and “Essencial”, its effect on the degree of inappropriateness was very limited. Although there are studies that point to the low value of performing PCXRs in patients with ASA grades I–II [25,43], these results are also obtained in other publications that do not consider this dimension [22], which is consistent with the results obtained in this study. This fact could explain that, with the exception of NH, the other initiatives evaluated do not consider this dimension in their recommendations; nor does DND, which considers ASA for the preoperative tests of “Resting ECG”, “Full blood count test” and “Kidney function test”, but not for PCXR [17].

However, the implementation of strict criteria for the indication of PCXR could be justified based on the results obtained in other studies. For example, in the year 2008, it was estimated that, in Spain, only 4.1% of the indicated CXRs found relevant findings [20], a value that decreased to 1.3% according to a meta-analysis of 1993 [21]; another study carried out in the United Kingdom determined that, in 1992, up to 75% of the PCXRs were not examined after their indication [39].

In turn, the knowledge and application of specific protocols for the indication of PCXRs have been shown to be effective in increasing the efficiency of a Health System [44]. This has also been studied in the indication of cranial radiographs, whose inappropriateness decreased from 25.1% to 14.7% through an intervention performed in 2001 in a Pediatric Emergency Service of a Spanish hospital [45].

With all this, and despite the high frequency of PCXRs in our environment, the extensive experience we have in their management and the possibility of improving the efficiency of the Healthcare System by reducing inappropriate PCXRs, there is still no clear consensus on when its indication is necessary. This situation already occurred before the emergence of the different initiatives mentioned, both in the indication of preoperative tests (including PCXRs) [46] and in the request for other radiological tests, such as simple radiography prescribed after a head injury [47].

Regarding the possible causes of this variability, it should be considered that these recommendations are usually prepared in collaboration with different scientific societies in the health field and that they are often the result of the application of a Delphi method [17,18,48]. Thus, ACR, SEDAR and SCARTD have participated in the elaboration of the recommendations of CW [16], NH [18] and “Essencial” [19], respectively; the NICE recommendations served as a source for the guidelines produced by the “European Society of Anaesthesiology” [23,49].

In addition, the recommendations present variations in their wording, which influence their precision and application likelihood. While NH expresses its recommendation in the negative “Do Not Perform PCXR in…” [18], CW and “Essencial” advise not to indicate them routinely and focus their content on the possible exceptions “It is a good idea to have a chest X-ray before you have surgery or before you go into the hospital if…” [16,19]. On the other hand, the NICE DND initiative also expresses its recommendation in the negative “Do not routinely offer chest X-rays before surgery” [17]; although the initial idea of the present study was to also evaluate the inappropriateness according to this recommendation, it could not finally be included because its lack of specificity in establishing what would be considered routine and what would not made its application and evaluation impossible.

Besides, despite the similarity between the CW and “Essencial” recommendations, this research included both initiatives to study a broader sample of international initiatives and to explore possible criteria and application differences between two recommendations aimed at a common geographical area (NH, as a recommendation issued by a Spanish institution; “Essencial”, as a recommendation published by a Catalan institution, a region of Spain).

Hence, the existence of different recommendations and protocols for the same healthcare procedure should raise the question of whether standardization is necessary, or if, on the contrary, this fact should be assumed as a variability of clinical practice, a consequence of the different population and organizational characteristics of each healthcare ordinance.

Moreover, each health institution should assess the different available recommendations and choose the one that best suits their Health System. For this, it must be considered that, although the assessed initiatives have international diffusion, all of them were developed with the collaboration of scientific societies from a particular geographical area. However, within the framework of the quality care programs, it could be interesting to introduce accurate, unequivocal and easily implemented and measurable recommendations. Thus, the compliance would be high, and its measurement would allow assessing its progression for establishing complementary improvement actions.

Furthermore, this wide variability between recommendations should be considered in the framework of medical education. Thus, it would be interesting if healthcare professionals received more specific training on initiatives and recommendations against the overuse of medical services, as it is shown that 64% of Spanish surgeons and anesthetists did not know the NH initiative in 2017 [3]. Additionally, this education could be complemented with assessment resources that allow professionals to critically analyze the different recommendations and clinical practice guidelines published by international institutions.

On the other hand, this study compared the degree of compliance with the three assessed recommendations, but it did not analyze the effect of each of them in real clinical practice, as it did not explore the impact or clinical benefits for the patients of the presence or absence of the indication of each PCXR. In this sense, it would be interesting to develop new studies that investigate the implications of each of these recommendations in the patient’s clinical progression and the Health System.

### 4.1. Cost Attributed to PCXR Overuse

The results of 2276.1 RVU/year (EUR 52,122.69) according to CW and “Essencial” and of 170.1 RVU/year (EUR 3895.29) according to NH, would be equivalent, for example, to the cost of 429 and 32 high definition chest computed tomography scans [34], respectively. However, the comparison of the cost derived from overuse presents many difficulties: on the one hand, the prices established for carrying out a CXR may vary between different territories and healthcare systems. For example, the Healthcare Service of the Basque Country (Osakidetza) estimates the cost of a PCXR during the year 2020 at EUR 22 or 0.94 RVU [35]; NICE, in the United Kingdom, values a PCXR at 29 pounds (including the expense derived from human resources and equipment) [17]. On the other hand, the few economic studies that have been carried out in this regard can also assign different prices to a PCXR and usually measure their results by units that are hardly comparable [26,30,38].

### 4.2. Limitations

The PCXRs, of the total CXRs, were detected and classified by recoding a free text field in which the requesting professional collected the reason for the indication. This would imply the impossibility of unequivocally identifying the reason for requesting the radiological test in those patients in whom it had not been explicitly recorded. To counteract this limitation, all CXRs indicated after the intervention were established as the exclusion criterion and only highly suggestive reasons for indication were classified as PCXRs, adopting a conservative attitude in recoding.

There was no explicit information on the presence of signs and symptoms of cardiopulmonary diseases, which did not allow us to explore the differences between CW and “Essencial” and could overestimate the degree of inappropriateness obtained for both initiatives. To mitigate this limitation conservatively, it was assumed that only patients with a DRG for thoracic intervention met these conditions, so the degree of actual inappropriateness could be greater.

Only the other previous CXRs performed in the study hospital itself and in associated Specialty Centers were assessed, as we were not able to include those carried out in other healthcare centers. This could underestimate the frequency of overprescription of CXRs due to duplication.

All the PCXRs indicated were counted according to the available electronic requests. Although some of these may not have been performed, the impact of this limitation would be small—barely 1.9% of the total, according to Flamm et al. [38].

The exact monetary equivalence of 1 radiological RVU of the Community of Madrid could not be applied, as it is not in the public domain. However, the price equivalent to the “Health Service of the Basque Country” (Osakidetza) was used. This is, organizationally, a very similar system, so the difference in the real cost is likely to be low. Finally, the real cost associated with inappropriateness could be higher, since it did not include costs derived from the indication of new cascade procedures as a consequence of the detection of “false positive” findings, although its precise estimation would be complex.

### 4.3. Strengths

To our knowledge, this is the first study that assesses the degree of inappropriateness in the indication of PCXRs, comparing the recommendations of CW, NH and “Essencial” as three institutions with wide international recognition. This allows it to be compared with other international studies, but also to assess the degree of follow-up of initiatives developed specifically for the Healthcare System in which this research was performed.

The methodology used allowed us to explore what variables are more associated with the inappropriateness of PCXRs and the costs associated with it, which are fundamental aspects to be able to develop measures that reduce overuse and, thereby, increase the efficiency of the Healthcare System. Thus, information campaigns should be promoted among health professionals, and the possibilities offered by health records and electronic request systems should be used to improve appropriateness, avoid duplication of PCXRs and avoid unnecessary risks for the patient.

Finally, this study represents an important advance in this line of work that allows the establishment of new scientific hypotheses and offers possible improvements in terms of quality of care, patient safety and the efficiency of the Healthcare System.

## 5. Conclusions

There is wide variability between the recommendations issued by institutions of recognized international prestige on the indication of PCXRs, which use different dimensions of the patient and the intervention to classify a PCXR as inappropriate.

This variability directly affects the degree of inappropriateness found. The factors that most affected a high degree of resulting inappropriateness were the existence of previous CXRs and not considering a PCXR appropriate when the intervention was to be performed on the chest.

The degree of inappropriateness obtained was highly variable depending on the nature of the intervention as “urgent” or “scheduled” and its administrative origin, but the assessment could be reversed depending on the recommendation applied. However, the indication of PCXRs by the Anesthesiology Department presented a lower degree of inappropriateness than the other petitioning surgical units.

The expense attributable to the inappropriateness of the PCXRs was highly variable depending on the recommendation applied, being EUR 52,122.69 (2276.1 RVU/year) according to CW and “Essencial” and EUR 3895.29 (170.1 RVU/year) according to NH.

The high percentage of inappropriateness observed according to some recommendations, together with the high frequency of indication of this test among the different healthcare systems, imply an opportunity cost with a significant margin for improvement. Disinvestment in this technology in situations where it is unnecessary, and its consequent reinvestment in other practices whose effectiveness and clinical benefit have been demonstrated, would enhance the sustainability of the Healthcare System, making it more efficient.

## Figures and Tables

**Figure 1 ijerph-17-08783-f001:**
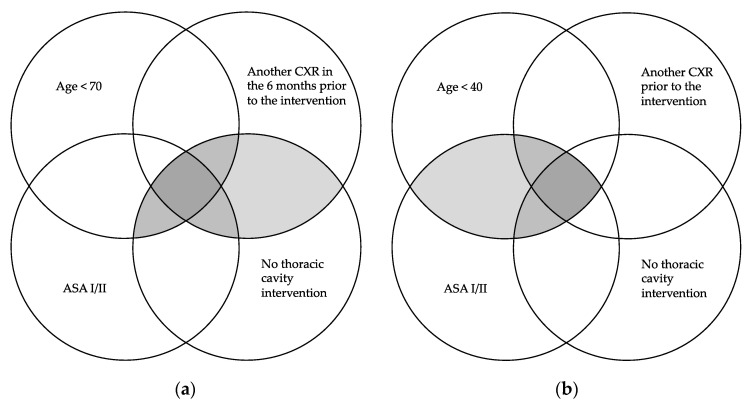
Inappropriateness assumptions according to the CW, NH and “Essencial” recommendations: (**a**) inappropriateness assumption for “Choosing Wisely” and “Essencial”; (**b**) Inappropriateness Assumption for “No Hacer”. CXR: chest X-ray. ASA: The American Society of Anesthesiologists Classification. The shaded areas represent the combinations of inappropriate indication of preoperative chest X-rays established by each recommendation.

**Table 1 ijerph-17-08783-t001:** Distribution of the characteristics of the interventions carried out between 1 July 2018 and 30 June 2019.

Characteristics of the Interventions	Global		Interventions without PCXR		Interventions with ≥1 PCXR		
	*n*	% (95% CI)	*n*	% (95% CI)	*n*	% (95% CI)	*p*
**Patient Gender**							
Women	14,551	52.2% (51.6%–52.8%)	13,097	52.4% (51.8%–53.1%)	1454	49.9% (48.1%–51.7%)	0.070
Men	13,339	47.8% (47.2%–48.4%)	11,881	47.6% (46.9%–48.2%)	1458	50.1% (48.3%–51.9%)	0.071
**Patient Age**							
Age: Median (Q1–Q3)		64 (48–76)		64 (48–76)		65 (53–75)	
≥40	23,365	83.8% (83.3%–84.2%)	20,712	82.9% (82.4%–83.4%)	2653	91.1% (90.0%–92.1%)	<0.001
≥60	16,085	57.7% (57.1%–58.3%)	14,253	57.1% (56.4%–57.7%)	1832	62.9% (61.1%–64.6%)	<0.001
≥70	11,108	39.8% (39.3%–40.4%)	9975	39.9% (39.3%–40.5%)	1133	38.9% (37.2%–40.7%)	0.503
**Patient ASA grade**							
ASA I or ASA II	14,768	71.7% (71.1%–72.4%)	13,458	73.3% (72.7%–73.9%)	1310	58.9% (56.9%–61.0%)	<0.001
ASA ≥ III	5816	28.3% (27.6%–28.9%)	4903	26.7% (26.1%–27.3%)	913	41.1% (39.0%–43.1%)	<0.001
**Department associated with the intervention**							
Ophthalmology	7078	25.4% (24.9%–25.9%)	6898	27.6% (27.1%–28.2%)	180	6.2% (5.4%–7.1%)	<0.001
Traumatology	4409	15.8% (15.4%–16.2%)	3835	15.4% (14.9%–15.8%)	574	19.7% (18.3%–21.2%)	0.008
General Surgery	4289	15.4% (15.0%–15.8%)	3822	15.3% (14.9%–15.8%)	467	16.0% (14.7%–17.4%)	0.677
Urology	2488	8.9% (8.6%–9.3%)	2178	8.7% (8.4%–9.1%)	310	10.6% (9.6%–11.8%)	0.267
Gynecology	1340	4.8% (4.6%–5.1%)	1178	4.7% (4.5%–5.0%)	162	5.6% (4.8%–6.5%)	0.637
Otolaryngology	1245	4.5% (4.2%–4.7%)	1111	4.4% (4.2%–4.7%)	134	4.6% (3.9%–5.4%)	0.935
Plastic Surgery	1220	4.4% (4.1%–4.6%)	1161	4.6% (4.4%–4.9%)	59	2.0% (1.6%–2.6%)	0.344
Vascular Surgery and Hemodynamics	1062	3.8% (3.6%–4.0%)	923	3.7% (3.5%–3.9%)	139	4.8% (4.1%–5.6%)	0.537
Maxillofacial surgery	998	3.6% (3.4%–3.8%)	908	3.6% (3.4%–3.9%)	90	3.1% (2.5%–3.8%)	0.791
Cardiac Surgery	785	2.8% (2.6%–3.0%)	499	2.0% (1.8%–2.2%)	286	9.8% (8.8%–11.0%)	<0.001
Neurological surgery	746	2.7% (2.5%–2.9%)	573	2.3% (2.1%–2.5%)	173	5.9% (5.1%–6.9%)	0.016
Dermatology	550	2.0% (1.8%–2.1%)	485	1.9% (1.8%–2.1%)	65	2.2% (1.8%–2.8%)	0.875
Pain + Palliative	517	1.9% (1.7%–2.0%)	504	2.0% (1.9%–2.2%)	13	0.4% (0.3%–0.8%)	0.688
Thoracic Surgery	482	1.7% (1.6%–1.9%)	332	1.3% (1.2%–1.5%)	150	5.2% (4.4%–6.0%)	0.013
Radiation oncology	327	1.2% (1.1%–1.3%)	227	0.9% (0.8%–1.0%)	100	3.4% (2.8%–4.2%)	0.102
Pediatric surgery	130	0.5% (0.4%–0.6%)	129	0.5% (0.4%–0.6%)	1	0.0% (0.0%–0.2%)	<0.001
Gastroenterology	103	0.4% (0.3%–0.4%)	96	0.4% (0.3%–0.5%)	7	0.2% (0.1%–0.5%)	0.952
Other	121	0.4% (0.4%–0.5%)	119	0.5% (0.4%–0.6%)	2	0.1% (0.0%–0.2%)	<0.001
**DRG intervention**							
Thoracic cavity intervention	1226	4.8% (4.6%–5.1%)	803	3.5% (3.3%–3.8%)	423	15.3% (14.0%–16.6%)	<0.001
No thoracic cavity intervention	24,234	95.2% (94.9%–95.4%)	21,885	96.5% (96.2%–96.7%)	2349	84.7% (83.4%–86.0%)	0.0000
**Intervention type**							
Outpatient	13,456	48.2% (47.7%–48.8%)	12,840	51.4% (50.8%–52.0%)	616	21.2% (19.7%–22.7%)	<0.001
Scheduled	10,914	39.1% (38.6%–39.7%)	8969	35.9% (35.3%–36.5%)	1945	66.8% (65.1%–68.5%)	<0.001
Urgent	3520	12.6% (12.2%–13.0%)	3169	12.7% (12.3%–13.1%)	351	12.1% (10.9%–13.3%)	0.730
**Intervention origin**							
Waiting List	20,418	73.2% (72.7%–73.7%)	18,649	74.7% (74.1%–75.2%)	1769	60.7% (59.0%–62.5%)	<0.001
Hospitalization	2931	10.5% (10.2%–10.9%)	2337	9.4% (9.0%–9.7%)	594	20.4% (19.0%–21.9%)	<0.001
Minor unscheduled surgery	2722	9.8% (9.4%–10.1%)	2268	9.1% (8.7%–9.4%)	454	15.6% (14.3%–17.0%)	<0.001
Emergencies	1819	6.5% (6.2%–6.8)	1724	6.9% (6.6%–7.2%)	95	3.3% (2.6%–4.0%)	0.168
**Other intervention in the previous 12 months**							
No	17,398	62.4% (61.8%–62.9%)	15,407	61.7% (61.1%–62.3%)	1991	68.4% (66.7%–70.0%)	<0.001
Yes	10,492	37.6% (37.1%–38.2%)	9571	38.3% (37.7%–38.9%)	921	31.6% (30.0%–33.3%)	<0.001
**Existence of another CXR performed** **in the months prior to the intervention**							
3 months	10,676	38.3% (37.7%–38.9%)	8281	33.2% (32.6%–33.7%)	2395	82.2% (80.8%–83.6%)	<0.001
6 months	12,956	46.5% (45.9%–47.0%)	10,314	41.3% (40.7%–41.9%)	2642	90.7% (89.6%–91.7%)	<0.001
9 months	14,134	50.7% (50.1%–51.3%)	11,420	45.7% (45.1%–46.3%)	2714	93.2% (92.2%–94.1%)	<0.001
12 months	15,059	54.0% (53.4%–54.6%)	12,325	49.3% (48.7%–50.0%)	2734	93.9% (93.0%–94.7%)	<0.001
**TOTAL**	27,890		24,978	89.6% (89.2%–89.9%)	2912	10.4% (10.1%–10.8%)	

CXR: chest X-ray; PCXR: preoperative chest X-ray; *n*: sample; % (95% CI): percentage (expected interval of such percentage with a confidence of 95%); *p*-value “*p*” of significance corresponding to the comparison of interventions without PCXR versus interventions ≥ 1 PCXR (hypothesis testing: parametric *Chi-square test* (*χ^2^*); non-parametric *Fisher’s exact test* if *n* < 5); Q1: quartile 1; Q3: quartile 3; ASA: The American Society of Anesthesiologists Classification; DRG: diagnostic related groups; sample reading: 90.7% of all interventions with at least one PCXR had another X-ray performed in the 6 months prior to the intervention. A total of 5.9% of all the interventions with at least one PCXR were associated with the Neurosurgery Service.

**Table 2 ijerph-17-08783-t002:** Degree of inappropriateness of the indication for preoperative chest X-rays, based on different assumptions of inappropriateness, and without considering whether the intervention was performed on the thoracic cavity as a criterion.

Degree of Inappropriateness		ASA I-II				ASA NC											
		AND				AND				AND				AND			
		<40 Years				<40 Years				<60 Years				<70 Years			
		iPCXR		Int. with iPCXR, of Total Int. in Which It Was Unnecessary		iPCXR		Int. with iPCXR, of Total Int. in Which It Was Unnecessary		iPCXR		Int. with iPCXR, of Total Int. in Which It Was Unnecessary		iPCXR		Int. with iPCXR, of Total Int. in Which It Was Unnecessary	
		*n*	% (95% CI)	*n*	% (95% CI)	*n*	% (95% CI)	*n*	%(95% CI)	*n*	% (95% CI)	*n*	% (95% CI)	*n*	% (95% CI)	*n*	% (95% CI)
OR	Prior CXR NC	187	5.4% (4.7%–6.2%) *	174	5.6% (4.8%–6.4%) *	288	8.4% (7.5%–9.3%)	259	5.7% (5.1%–6.4%)	1212	35.1% (33.6%–36.8%)	1080	9.1% (8.6%–9.7%)	2079	60.3% (58.6%–61.9%)	1779	10.6% (10.1%–11.1%)
	Other CXR in 3 months prior–int.	2946	85.4% (84.2%–86.6%)	2434	18.5% (17.8%–19.2%)	2956	85.7% (84.5%–86.8%)	2444	17.2% (16.6%–17.8%)	3128	90.7% (89.7%–91.6%)	2610	14.0% (13.5%–14.5%)	3246	94.1% (93.3%–94.9%)	2722	12.7% (12.3%–13.2%)
	Other CXR in 6 months prior–int.	3187	92.4% (91.5%–93.2%)	2664	17.5% (16.9%–18.1%)	3194	92.6% (91.7%–93.4%)	2671	16.4% (15.8%–17.0%)	3288	95.3% (94.6%–96.0%)	2761	13.7% (13.3%–14.2%)	3346	97.0% (96.4%–97.5%)	2816	12.6% (12.2%–13.1%)
	Other CXR in 9 months prior–int.	3263	94.6% (93.8%–95.3%)	2731	16.7% (16.1%–17.3%)	3270	94.8% (94.0%–95.5%)	2738	15.8% (15.2%–16.3%)	3337	96.8% (96.1%–97.3%)	2804	13.4% (13.0%–13.9%)	3371	97.7% (97.2%–98.2%)	2838	12.4% (12.0%–12.9%)
	Other CXR in 12 months prior–int.	3282	95.2% (94.4%–95.8%)	2750	15.9% (15.4%–16.5%)	3289	95.4% (94.6%–96.0%)	2757	15.1% (14.6%–15.7%)	3349	97.1% (96.5%–97.6%)	2816	13.1% (12.7%–13.6%)	3381	98.0% (97.5%–98.4%)	2848	12.2% (11.8%–12.6%)

CXR: chest X-ray; iPCXR: unnecessarily indicated preoperative chest X-ray; *n*: sample; % (95% CI): percentage (expected interval of such percentage with a confidence of 95%); NC: Not considered; ASA: The American Society of Anesthesiologists Classification; Int: surgical intervention; * result of inappropriateness according to the assumptions recommended by the “No Hacer” initiative. The denominator for calculating each percentage iPCXR is the same in all cases, since it was calculated using the total of 3449 PCXRs indicated in the period studied. The denominator for calculating the percentage of interventions with at least one PCXR, out of the total number of interventions in which this indication was unnecessary, was variable, since the different combinations of assumptions encompass different sets of interventions within the same sample; reading example: if we consider the PCXR indication inappropriate in: (1) ASA I or ASA II patients, under 40 years of age; (2) patients who have already undergone another X-ray in the 9 months prior to the evaluated intervention; then 94.6% of the PCXRs were iPCXRs. Reading example: if we consider the PCXR indication inappropriate in: (1) ASA I or ASA II patients, under 40; (2) patients who have already undergone another X-ray in the 9 months prior to the evaluated intervention; then at least one PCXR was indicated in 16.7% of the interventions in which this indication was unnecessary.

**Table 3 ijerph-17-08783-t003:** Degree of inappropriateness of the indication for preoperative chest X-rays, based on different assumptions of inappropriateness, considering any preoperative chest X-ray as appropriate that was indicated for a thoracic intervention, regardless of age and the existence of another previous chest X-ray.

Degree of Inappropriateness		ASA I–II				ASA NC											
		AND				AND				AND				AND			
		<40 Years				<40 Years				<60 Years				<70 Years			
		IPCXR, of the Total PCXR Performed		Int. with iPCXR, of Total Int. in Which It Was Unnecessary		IPCXR, of the Total PCXR Performed		Int. with iPCXR, of Total Int. in Which It Was Unnecessary		IPCXR, of the Total PCXR Performed		Int. with iPCXR, of Total Int. in Which It Was Unnecessary		IPCXR, of the Total PCXR Performed		Int. with iPCXR, of Total Int. in Which It Was Unnecessary	
		*n*	% (95% CI)	*n*	% (95% CI)	*n*	% (95% CI)	*n*	% (95% CI)	*n*	% (95% CI)	*n*	% (95% CI)	*n*	% (95% CI)	*n*	% (95% CI)
OR	Prior CXR NC	162	4.7% (4.0%–5.5%)	149	5.7% (4.9%–6.6%)	227	6.6% (5.8%–7.5%)	207	5.5% (4.8%–6.3%)	976	28.3% (26.8%–29.8%)	909	9.0% (8.5%–9.6%)	1612	46.7% (45.1%–48.4%)	1440	10.0% (9.5%–10.5%)
	Other CXR in 3 months prior–int.	2152	62.4% (60.8%–64.0%)	1898	17.8% (17.1%–18.5%)	2162	62.7% (61.1%–64.3%)	1908	16.5% (15.8%–17.2%)	2327	67.5% (65.9%–69.0%)	2067	13.2% (12.7%–13.7%)	2440	70.7% (69.2%–72.2%)	2174	11.9% (11.5%–12.4%)
	Other CXR in 6 months prior–int.	2378	68.9% (67.4%–70.5%)	2113	16.7% (16.1%–17.4%)	2385	69.2% (67.6%–70.7%)	2120	15.7% (15.1%–16.3%)	2474	71.7% (70.2%–73.2%)	2205	13.0% (12.5%–13.5%)	2529	73.3% (71.8%–74.8%) *	2257	11.8% (11.4%–12.3%) *
	Other CXR in 9 months prior–int.	2450	71.0% (69.5%–72.5%)	2176	15.9% (15.3%–16.5%)	2457	71.2% (69.7%–72.7%)	2183	15.0% (14.4%–15.6%)	2520	73.1% (71.6%–74.5%)	2245	12.7% (12.2%–13.2%)	2553	74.0% (72.5%–75.5%)	2278	11.6% (11.2%–12.1%)
	Other CXR in 12 months prior–int.	2468	71.6% (70.0%–73.0%)	2194	15.1% (14.5%–15.7%)	2475	71.8% (70.2%–73.2%)	2201	14.3% (13.8%–14.9%)	2531	73.4% (71.9%–74.8%)	2256	12.4% (11.9%–12.8%)	2563	74.3% (72.8%–75.7%)	2288	11.4% (11.0%–11.9%)

CXR: chest X-ray; iPCXR: unnecessarily indicated preoperative chest X-ray; n: sample; % (95% CI): percentage (expected interval of such percentage with a confidence of 95%); NC: Not considered; ASA: The American Society of Anesthesiologists Classification. Int: surgical intervention; * result of inappropriateness according to the assumptions recommended by the “Choosing Wisely” and “Essencial” initiatives; the denominator for calculating each percentage iPCXR is the same in all cases, since it was calculated on the total of 3449 PCXRs indicated in the period studied. The denominator for calculating the percentage of interventions with at least one PCXR, out of the total number of interventions in which this indication was unnecessary, is variable, since the different combinations of assumptions encompass different sets of interventions within the same sample; reading example: if we consider the PCXR indication inappropriate in: (1) ASA I or ASA II patients, under 40 years of age and who do not present cardiopulmonary pathology; (2) patients who have already undergone another X-ray in the 9 months prior to the evaluated intervention and who do not present cardiopulmonary pathology; then 71.0% of the PCXRs were iPCXRs. Reading example: if we consider the PCXR indication inappropriate in: (1) ASA I or ASA II patients, under 40 and who do not present cardiopulmonary pathology; (2) patients who have already undergone another X-ray in the 9 months prior to the evaluated intervention and who do not present cardiopulmonary pathology; then at least one PCXR was indicated in 15.9% of the interventions in which this indication was unnecessary.

**Table 4 ijerph-17-08783-t004:** Degree of inappropriateness of the indication for preoperative chest X-rays, according to the recommendations of “Choosing Wisely”, “Essencial” and “No Hacer”, stratified by characteristics of the interventions.

Characteristics of the Interventions	“Choosing Wisely” and “Essencial”			“No Hacer”		
	*n*	% (95% CI)	*p*	*n*	% (95% CI)	*p*
**Intervention Origin**						
Waiting List	1414	9.6% (9.1%–10.1%)	<0.001	125	5.1% (4.3%–6.0%)	<0.001
Hospitalization	416	19.0% (17.4%–20.7%)		11	9.1% (5.1%–15.7%)	
Minor unscheduled surgery	385	20.0% (18.3%–21.9%)		29	13.4% (9.5%–18.7%)	
Emergencies	42	19.8% (15.0%–25.7%)		9	2.66% (1.4%–5.0%)	
**Intervention type**						
Non-urgent	2073	11.6% (11.2%–12.1%)	0.001	165	6.1% (5.2%–7.0%)	0.002
Urgent	184	14.7% (12.9%–16.8%)		9	2.2% (1.2%–4.2%)	
**Existence of another CXR performed in the 3 months prior to the intervention**						
No	394	3.7% (3.4%–4.1%)	<0.001	39	1.6% (1.2%–2.1%)	<0.001
Yes	1863	21.8% (21.0%–22.7%)		135	20.6% (17.7%–23.9%)	
**Existence of another CXR performed in the 6 months prior to the intervention**						
No	163	1.9% (1.7%–2.3%)	<0.001	22	1.0% (0.6%–1.4%)	<0.001
Yes	2094	19.6% (18.9%–20.4%)		152	18.5% (16.0%–21.3%)	
**Existence of another CXR performed in the 9 months prior to the intervention**						
No	116	1.5% (1.2%–1.8%)	<0.001	17	0.8% (0.5%–1.2%)	<0.001
Yes	2141	19.0% (18.2%–19.7%)		157	17.7% (15.3%–20.3%)	
**Existence of another CXR performed in the 12 months prior to the intervention**						
No	107	1.5% (1.2%–1.8%)	<0.001	16	0.7% (0.4%–1.2%)	<0.001
Yes	2150	18.3% (17.6%–19.0%)		158	16.7% (14.4%–19.2%)	
**Other intervention performed in the previous 12 months**						
No	1519	12.2% (11.6%–12.8%)	0.028	136	5.2% (4.4%–6.2%)	0.093
Yes	738	11.1% (10.4%–11.9%)		38	7.1% (5.2%–9.6%)	
**TOTAL**	2257	11.8% (11.4%–12.3%)		174	5.6% (4.8%–6.4%)	

CXR: chest X-ray; *n*: sample; % (95% CI): percentage (expected interval of such percentage with a confidence of 95%); *p*: *p*-value of significance (hypothesis testing: parametric Chi-square test (χ^2^)); this table expresses the percentage of interventions in which at least one preoperative chest X-ray (PCXR) was indicated, calculated out of the total number of interventions in which such indication was unnecessary, and based on each characteristic assessed; sample reading: at least one PCXR was indicated in 14.7% of the “urgent” interventions in which such an indication was unnecessary according to the recommendations of “*Choosing Wisely*” and “Essencial”, compared to 11.6% of the “non-urgent” cases. This trend changed according to the recommendation “No Hacer”, according to which at least one PCXR was indicated in 2.2% of the “urgent” interventions in which such an indication was unnecessary, compared to 6.1% of the “non-urgent”.

**Table 5 ijerph-17-08783-t005:** Degree of inappropriateness of the indication for preoperative chest X-rays, according to the recommendations of “Choosing Wisely”, “Essencial” and “No Hacer”, stratified by characteristics of the indications of preoperative chest X-rays.

Characteristics of the Indications	“Choosing Wisely” and “Essencial’			“No Hacer”		
	*n*	% (95% CI)	*p*	*n*	% (95% CI)	*p*
**CXR requesting department**						
Anesthesiology	280	57.5% (53.1%–61.8%)	<0.001	20	4.1% (2.7%–6.3%)	<0.001
Cardiac surg.	10	3.5% (1.9%–6.4%)		2	0.7% (0.2%–2.8%)	
General surg.	322	93.9% (90.8%–96.0%)		28	8.2% (5.7%–11.6%)	
Pediatric surg.	1	100.0% (2.5%–100.0%)		1	100.0% (2.5%–100.0%)	
Plastic surg.	34	91.9% (77.4%–97.4%)		3	8.1% (2.6%–22.6%)	
Chest surg.	22	15.4% (10.3%–22.3%)		7	4.9% (2.3%–9.9%)	
Dermatology	57	78.1% (67.1%–86.2%)		0	0.0% (0.0%–4.9%)	
Gynecology	157	94.0% (89.2%–96.8%)		3	1.8% (0.6%–5.4%)	
Vascular surg. and Hemodynamics	68	91.9% (83.0%–96.3%)		1	1.4% (0.2%–9.1%)	
Maxillofacial surg.	69	92.0% (83.2%–96.4%)		17	22.7% (14.5%–33.6%)	
Neurosurgery	160	88.4% (82.8%–92.3%)		10	5.5% (3.0%–10.0%)	
Ophthalmology	87	73.7% (65.0%–80.9%)		13	11.0% (6.5%–18.1%)	
Otorhinolaryngology	99	79.8% (71.8%–86.0%)		20	16.1% (10.6%–23.7%)	
Traumatology	481	90.9% (88.2%–93.1%)		26	4.9% (3.4%–7.1%)	
Urology	263	90.1% (86.1%–93.0%)		8	2.7% (1.4%–5.4%)	
Emergencies	109	86.5% (79.3%–91.5%)		13	10.3% (6.1%–17.0%)	
Medical and central services	310	78.9 (74.5%–82.8%)		15	3.8% (2.2%–6.2%)	
**Requesting service identical to service associated with surgery**						
No	737	66.7% (63.9%–69.4%)	<0.001	51	4.6% (3.5%–6.0%)	0.151
Yes	1792	76.5% (74.7%–78.1%)		136	5.8% (4.9%–6.8%)	
**TOTAL**	2529	73.3% (71.8%–74.8%)		187	5.4% (4.7%–6.2%)	

*n*: sample; % (95% CI): percentage (expected interval of such percentage with a confidence of 95%). *p*: *p*-value of significance (hypothesis testing: parametric Chi-square test (χ^2^); non-parametric Fisher’s exact test if n < 5); CXR: chest X-ray; Surg.: surgery; sample reading: According to “Choosing Wisely” and “Essencial”, 73.7% of the preoperative chest X-rays indicated by the Ophthalmology Service were unnecessary, compared to 3.5% of those indicated by Cardiac Surgery.

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
