# Peer review of "Estimation of the Overuse of Preoperative Chest X-rays According to “Choosing Wisely”, “No Hacer”, and “Essencial” Initiatives: Are They Equally Applicable and Comparable?"

_ijerph, 2020, doi:10.3390/ijerph17238783_

Round 1

Reviewer 1 Report

The manuscript is according to the audience of the IJERPH.

The internal consistency is a strength of this manuscript.

The use of English and style is very good. Just minor suggestion, for instance in line 171 "2) related to the intervention"

Author Response

**Response is also available in the attached file.**

Dear Reviewer,

Thank you so much for your comments; we appreciate your interest in our work and your suggestions, which help us to improve our manuscript.

Please, find below our answers for each of your observations (in blue color, the reviewer’s comments; in red color, the author’s responses):

Point 1.  The use of English and style is very good. Just minor suggestion, for instance in line 171 "2) related to the intervention". 

Response 1. Thank you very much for this suggestion, we agree the change proposed will make the manuscript more understandable. We have made the following modifications:

  • Line 188: “Related to the patient:”
  • Line 189: “Related to the intervention:”
  • Line 191: “Related to the PCXR:”

We enclosed a marked copy of the manuscript, including all of these changes. Please, let us know if you find them suitable, or if you consider other changes to enhance the paper.

Thank you so much.

Best regards.

Reviewer 2 Report

Thank you for submitting your research manuscript on the overuse of pre-operative chest x-rays. This study examined the inappropriate use of prescribing pre-operative chest x-rays using recommendations from three institutions with international recognition. I believe the study contributes to the existing body of literature with its international scope and by investigating an important problem in health care today: that of inappropriate use of medical tests and procedures. The findings are clearly explained, including the variability in findings based on the international standards applied for determining whether the procedure is appropriate. I noticed on line 360 the statement is made that the study "allowed us to assess main causes" of overuse. I do not believe that causal conclusions can be drawn from this study since it is not an experimental design. Also, could you define what is meant by a relative value unit and explain the ASA I and II criteria is a little more depth? It is unclear to me as someone who is not an expert in this field. Also, I would suggest expanding your discussion of recommendations and implications to include implications for medical education of practitioners on inappropriate overuse of procedures.

Author Response

**Response is also available in the attached file.**

Dear Reviewer,

Thank you so much for your comments; we appreciate your interest in our work and your suggestions, which help us to improve our manuscript.

Please, find below our answers for each of your observations (in blue color, the reviewer’s comments; in red color, the author’s responses):

Point 1.  I noticed on line 360 the statement is made that the study "allowed us to assess main causes" of overuse. I do not believe that causal conclusions can be drawn from this study since it is not an experimental design.

Response 1. Thank you very much, we agree with your suggestion. Therefore, we have changed this affirmation, avoiding the causal conclusion and providing the explored associations. Besides, we have reviewed the whole text looking for similar assumptions in order to correct them. Thus, we have made the following modifications:

  • Line 416: “The methodology used allowed us to explore what variables are more associated with the inappropriateness of PCXR and the cost associated with it”
  • Line 275: “Performing another preoperative CXR in the months prior to the intervention constitutes the assumption associated with a greater increase in inappropriateness:”

Point 2.  Also, could you define what is meant by a relative value unit and explain the ASA I and II criteria is a little more depth? It is unclear to me as someone who is not an expert in this field.

Response 2. Thank you very much, we agree that these terms could be unknown for professionals of other research fields. Therefore, we have included the following explanations, which also have a specific bibliographic reference for further details:

  • Lines 179 to 183: “To estimate the healthcare cost corresponding to the iPCXR, Relative Value Units (RVU) was used as a measuring tool that integrates the personnel cost, maintenance cost, depreciation costs, and other charges directly related to the performance of a healthcare procedure[32]. Thus, this study used the value of 0.9 RVU per iPCXR, stipulated by the Community of Madrid[33] and the ‘Spanish Society of Medical Radiology’[34].”

Reference included: Baadh, A.; Peterkin, Y.; Wegener, M.; Flug, J.; Katz, D.; Hoffmann, J.C. The Relative Value Unit: History, Current Use, and Controversies. Curr Probl Diagn Radiol 2016, 45, 128–132, doi:10.1067/j.cpradiol.2015.09.006.

  • Line 127 to 129: “The MSSSI, in collaboration with the ‘Spanish Society of Anaesthesiology, Resuscitation and Pain Therapeutics’ (SEDAR), included ‘The American Society of Anesthesiologists (ASA)’ physical status classification system[31] in its evaluation, and established the following recommendation through the NH initiative: "do not perform preoperative chest radiography in patients under 40 years of age with physical status ASA I or II"[18]. In this respect, The ASA physical status classification system includes “healthy patients” as ASA I and “patients with mild systemic disease” as ASA II, while an ASA grade ≥ III implies a “severe systemic disease” or worse health-conditions[31].”

Reference included: American Society of Anesthesiologists ASA Physical Status Classification System Available online: https://www.asahq.org/standards-and-guidelines/asa-physical-status-classification-system (accessed on Jul 13, 2020).

Point 3.  Also, I would suggest expanding your discussion of recommendations and implications to include implications for medical education of practitioners on inappropriate overuse of procedures.

Response 3. Thank you very much for your comment, we also think that including potential implications for the medical training would improve the final version of the manuscript. Therefore, we have added the following paragraph in the “Discussion Section”:

  • Lines 361 to 367: “Furthermore, this wide variability between recommendations should be considered in the framework of medical education. Thus, it would be interesting that healthcare professionals receive more specific training on initiatives and recommendations against the overuse of medical services, as shows that 64% of Spanish surgeons and anesthetists would not know the NH initiative[3]. Also, this education could be complemented with assessment resources that allow professionals to critically analyze the different recommendations and clinical practice guidelines published by international institutions.”

We enclosed a marked copy of the manuscript, including all of these changes. Please, let us know if you find them suitable, or if you consider other changes to enhance the paper.

Thank you so much.

Best regards.

Reviewer 3 Report

Firstly, I thank the Editorial Committee for the opportunity to review this manuscript. The authors present a relevant study for clinical practice, contributing to a better understanding of the inappropriateness of the indication for preoperative chest X-rays (PCXR), and comparing the recommendations of 3 institutions with wide international recognition.  Furthermore, they analyze adequately the economic impact of the results obtained. The proposed manuscript meets adequately the purposes of the journal.

However, some recommendations are suggested below in order to improve the quality of the manuscript for its publication.

In the Abstract section, it should be advisable to include the location of the tertiary hospital (region and country), where the study was conducted. Furthermore, it would be desirable to include the total number of PCXR performed during this study (3,449), in order to estimate the magnitude of the percentages shown.

The Introduction section is well carried out and shows adequately the research status and problem.

In the Material and Methods section, it should be advisable to include the hospital bed capacity to estimate its size and volume of possible PCXR. The inappropriateness assumptions of the indication for PCXR based on the Choosing Wisely and Essencial initiative recommendations are quite similar. It would be desirable to comment on this issue in the Selected recommendations subsection, showing the possible differences between the two initiatives. Why were included the two initiatives in the study, if they are quite similar? The acronym for Choosing Wisely is incorrect in line 151 and 154, please review. Figure 1 is quite illustrative and informative.

Although the Tables are too long, the Results section is well carried out. However, the reader could think the Choosing Wisely and Essencial initiative recommendations are equal, so this issue should be clarified in the previous sections.

In the Discussion section, I consider the wide variations in the degree of the inappropriateness of PCXR obtained using the 3 initiatives should be explained in the first paragraph of this section. How can be explained briefly these differences? Maybe these differences are explained in the following paragraphs, but it should be desirable to explain or present them shortly in the first paragraph. Since the wide recommendation variety, it should be advisable the authors include some recommendations for clinical practice, indicating which initiative they recommend and in which clinical situations, based on the results obtained. Which initiative indicates better the need for a PCXR? Should it be recommended to include compulsorily any initiative in preoperative protocols?  

Finally, the manuscript is correctly referenced and references are updated (47% of references correspond to the last 5 years, while 70% to the last 10 years).

Author Response

**Response is also available in the attached file.**

Dear Reviewer,

Thank you so much for your comments; we appreciate your interest in our work and your suggestions, which help us to improve our manuscript.

Please, find below our answers for each of your observations (in blue color, the reviewer’s comments; in red color, the author’s responses):

Point 1.  In the Abstract section, it should be advisable to include the location of the tertiary hospital (region and country), where the study was conducted. Furthermore, it would be desirable to include the total number of PCXR performed during this study (3,449), in order to estimate the magnitude of the percentages shown.

Response 1. Of course, we also think that it is necessary information for the proper interpretation of the results included in the Abstract. Thank you very much for this wise suggestion. Thus, we have made the following changes:

  • Line 33: “Descriptive observational study with analytical components performed in a tertiary hospital in the Community of Madrid (Spain), between July 2018 and June 2019.”
  • Lines 36 to 37: “3,449 preoperative chest X-ray tests were performed during the period of study. 5.4% of them were unjustified according to the ‘No Hacer’ recommendation and 73.3% according to ‘Choosing Wisely’ and ‘Essencial’ criteria.”

Point 2.  In the Material and Methods section, it should be advisable to include the hospital bed capacity to estimate its size and volume of possible PCXR.

Response 2. Thank you very much for your comment, it would be interesting information for this section that, without doubt, it will improve the ‘Methods’ section. Therefore, we have included the hospital capacity of beds and operating rooms:

  • Lines 102 to 103: “Descriptive observational study with analytical components performed through the Infomation Systems records of a tertiary hospital in the Community of Madrid (Spain) with a capacity of 901 beds and 45 operating rooms”

Point 3.  The inappropriateness assumptions of the indication for PCXR based on the Choosing Wisely and Essencial initiative recommendations are quite similar. It would be desirable to comment on this issue in the Selected recommendations subsection, showing the possible differences between the two initiatives.

Response 3. Thank you so much for your suggestion. Effectively, both Choosing Wisely and Essencial initiative recommendations are very similar. The differences between them are subtle and barely affect the real practice. However, we agree with you that to provide the main differences between these recommendations will make the manuscript more understandable. Therefore, we have included the following paragraph in the ‘Selected Recommendations’ section:

  • Lines 138 to 143: “Thus, CW and ‘Essencial’ initiatives can be considered very similar recommendations that barely differ in small details such as 1) CW specifies the possible symptoms of heart or lung condition. 2) CW considers appropriate the PCXR indicated in patients with heart or lung condition, regardless of the presence of symptoms. 3) Essencial indicates PCXR to patients older than 70, with stable chronic cardiopulmonary disease, and without previous CXR in the last 6 months; while CW does not consider necessary the existence of stable chronic cardiopulmonary disease for this indication.”

Point 4.  Why were included the two initiatives in the study, if they are quite similar?.

Response 4. Thank you very much for your interest in this issue. We considered it interesting to include CW and Essencial in this study because it would allow us to explore a broader sample of initiatives in this framework. In this regard, we also considered to analyze the ‘Do Not Do’ recommendation published by The National Institute for Health and Care Excellence (NICE), but, as we explain in the ‘Discussion Section’, it was finally not possible because of its lack of specificity in the redaction.

In addition, to include the Essencial initiative allowed us to explore the differences between two recommendations published by related institutions and aimed to the same geographical territory: NH to the whole Spanish area, and Essencial to the Catalonian Health System, which belongs to the Spanish country. This explanation has also been included in the manuscript, in the “Discussion” section.

  • Lines 343 to 347: Besides, despite the similarity between the CW and Essencial recommendations, this research included both initiatives to study a broader sample of international initiatives and to explore possible criteria and application differences between two recommendations aimed at a common geographical area (NH, as a recommendation issued by a Spanish institution; and ‘Essencial’, as a recommendation published by a Catalan institution, a region of the Spanish country).”

Point 5.  The acronym for Choosing Wisely is incorrect in line 151 and 154, please review.

Response 5. Thank you so much for this advice, we have changed it to the right way in the new version. We appreciate this annotation.

  • Line 167: “The inappropriateness assumptions of the CW, NH and ‘Essencial…’”
  • Line 170: “Figure 1. Inappropriateness Assumptions according to the CW, NH and ‘Essencial…’”

Point 6.  However, the reader could think the Choosing Wisely and Essencial initiative recommendations are equal, so this issue should be clarified in the previous sections.

Response 6. Thank you very much again for your suggestion. We could observe in the response Nº4 the differences between Choosing Wisely and Essencial. However, we agree that it would be necessary to explain why in the “Results” Section both recommendations are shown together.

The reason for this presentation is because, although slight differences between Choosing Wisely and Essencial (like the presence of symptoms or a stable chronic cardiopulmonary disease) exist, they could not be analyzed because the quality of the records was not higher enough. We have added this information in the manuscript in the “Analysis Plan” subsection of the “Methodology” section:

  • Lines 149 to 158: “The records did not provide information about chronic disease, signs, or symptoms. For this reason, to assess inappropriateness according to the recommendations of the CW and ‘Essencial’ initiatives, it was considered that those patients whose intervention had a Diagnosis Related Group (DRG) code of surgery that compromised the thoracic cavity had signs, symptoms, or a diagnosis of heart or pulmonary disease and they would have been indicated for PCXR correctly. However, it was not possible to identify patients with these conditions but without a thoracic cavity surgery DRG code. Therefore, the quality of the records did not allow us to explore the subtle differences existing between CW and ‘Essencial’ about the presence of chronic disease, signs, or symptoms. Thus, both recommendations were finally assessed under the same criteria, and their results are shown together in the next sections of the manuscript.”

Also, we have included this aspect more specifically in the “Limitations” subsection of the “Discussion” section:

  • Lines 392 to 394: “There was no explicit information on the presence of signs and symptoms of cardiopulmonary diseases, which did not allow us to explore the differences between CW and ‘Essencial’ and could overestimate the degree of inappropriateness obtained for both initiatives.”

Point 7.  In the Discussion section, I consider the wide variations in the degree of the inappropriateness of PCXR obtained using the 3 initiatives should be explained in the first paragraph of this section. How can be explained briefly these differences? Maybe these differences are explained in the following paragraphs, but it should be desirable to explain or present them shortly in the first paragraph.

Response 7. We appreciate your comment. It would be a good point to improve the “Discussion” section. Following your recommendation, we have included a new first paragraph in this section:

  • Lines 254 to 263: “Wide variations were found in the degree of inappropriateness of PCXR according to the recommendations assessed (5.4% according to ‘No Hacer’ recommendation, and 73.3% according to ‘Choosing Wisely’ and ‘Essencial’ criteria). The performance of another CXR in the months prior to the intervention showed a high association for inappropriateness of PCXR, although the young age of the patient and the absence of a thoracic cavity surgery DRG would also increase overuse. Unnecessary PCXR were more indicated in ASA ≥ III patients according to CW and ‘Essencial’; and the Services of Maxillofacial Surgery, Otorhinolaryngology, or Ophthalmology also indicated more unnecessary PCXR than Cardiac Surgery, Thoracic Surgery, or Anesthesiology. The cost associated with overuse shows a high variability between 52,122.69 and 3,895.29 euros/year, depending on the recommendation applied.”

Point 8.  Since the wide recommendation variety, it should be advisable the authors include some recommendations for clinical practice, indicating which initiative they recommend and in which clinical situations, based on the results obtained. Which initiative indicates better the need for a PCXR? Should it be recommended to include compulsorily any initiative in preoperative protocols?.

Response 8. Thank you for your comment. That is an interesting point about the practical implications of the research. The authors think that each organization has to value the different recommendations and choose the one which fits more adequately with its Health Service. Besides, it would be interesting to implement unequivocal recommendations to achieve a good grade of compliance. Also, it is significant to note that this study compared the degree of compliance with the three assessed recommendations, but it did not analyze the effect of each of them in real clinical practice. Therefore, it would be necessary to develop other investigations to determine which recommendation is more suitable for each Health System.

We have included these points in the manuscript, within the “Discussion” section:

  • Lines 353 to 360: “Moreover, each health institution should assess the different available recommendations and choose the one that best suits their Health System. For that, it must be considered that, although the assessed initiatives have international diffusion, all of them were developed with the collaboration of scientific societies from a particular geographical area. However, within the framework of the quality care programs, it could be interesting to introduce accurate, unequivocal, and easily implemented and measurable recommendations. Thus, the compliance would be high, and its measurement would allow assessing its progression for establishing complementary improvement actions.“
  • Lines 368 to 373: “In the other hand, this study compared the degree of compliance with the three assessed recommendations, but it did not analyze the effect of each of them in real clinical practice, as it did not explore the impact or clinical benefit for the patients of the presence or absence of the indication of each PCXR. In this sense, it would be interesting to develop new studies that investigate the implications of each of these recommendations in the patient's clinical progression and the Health System.”

We enclosed a marked copy of the manuscript, including all of these changes. Please, let us know if you find them suitable, or if you consider other changes to enhance the paper.

Thank you so much.

Best regards.
